# FLASH-MM: fast and scalable single-cell differential expression analysis using linear mixed-effects models

Changjiang Xu[1,7], Delaram Pouyabahar [1,2,7], Veronique Voisin [3], Hamed Heydari [1,2] & Gary D. Bader [1,2,3,4,5,6] ✉

Single-cell RNA sequencing (scRNA-seq) enables detailed comparisons of gene expression across cells and conditions. Single-cell differential expression analysis faces challenges like sample correlation, individual variation, and scalability. We develop a fast and scalable linear mixed-effects model (LMM) estimation algorithm, FLASH-MM, to address these issues. We reformulate aspects of the linear mixed model estimation procedure to make it faster, by reducing computational complexity and memory usage. Simulation studies with scRNA-seq data show that FLASH-MM is accurate, computationally efficient, effectively controls false positive rates, and maintains high statistical power in differential expression analysis. Tests on tuberculosis immune and kidney single cell data demonstrate FLASH-MM's utility in accelerating single-cell differential expression analysis across diverse biological contexts.

Differential expression (DE) analysis is a cornerstone of transcriptomics research. Single-cell RNA sequencing (scRNA-seq) technology enables researchers to profile the transcriptomes of individual cells, uncovering transcriptional similarities and differences across various biological conditions for specific cell types. Advancements in cost efficiency and throughput have facilitated the generation of large-scale datasets comprising hundreds of subjects and millions of cells, opening new avenues for exploring cellular heterogeneity and dynamics across diverse conditions and samples. Cells from the same individual share common genetic and environmental backgrounds, resulting in a hierarchical structure and statistical dependencies among individual cells in scRNA-seq data[1]. This introduces significant challenges for differential expression analysis in single-cell studies, particularly due to the correlation within cell populations of each subject (intra-subject correlation[1]) and the high variability across cell populations from different subjects (inter-subject variability[2]). Ignoring these correlations and variabilities can inflate false positive rates in statistical tests[1,2]. Furthermore, the large scale of single-cell data, often encompassing hundreds of thousands to millions of cells, adds computational complexity, requiring efficient methods to manage and analyze these vast datasets effectively.

Linear mixed-effects models (LMMs) provide a framework to address the challenges of intra-subject correlation and inter-subject variability in single-cell differential expression analysis by incorporating fixed effects, which capture systematic differences across experimental conditions, and random effects, which model the correlations within subjects and the variations between subjects[1,3–5]. In multi-subject single-cell studies, cells are nested within subjects, and subjects are often nested within experimental conditions, meaning that cells from the same subject are correlated, and subjects within the same condition share common sources of variation. To account for this hierarchical structure, instead of modeling subjects as fixed effects, mixed models treat them as random effects, efficiently capturing both within-subject correlation and between-subject variability. Many methods and software packages have been developed to fit mixed-effects models[5–10]. The most widely used package is lme4[6], which uses maximum likelihood[11] or restricted maximum likelihood[12] methods for model fitting. However, fitting the LMM is

[1]The Donnelly Centre, University of Toronto, Toronto, Ontario, Canada. [2]Department of Molecular Genetics, University of Toronto, Toronto, Ontario, Canada. [3]Princess Margaret Research Institute, University Health Network, Toronto, Ontario, Canada. [4]Department of Computer Science, University of Toronto, Toronto, Ontario, Canada. [5]Lunenfeld-Tanenbaum Research Institute, Toronto, Ontario, Canada. [6]CIFAR Multiscale Human Program, CIFAR, Toronto, Ontario, Canada. [7]These authors contributed equally: Changjiang Xu, Delaram Pouyabahar. ✉e-mail: gary.bader@utoronto.ca

computationally demanding, particularly in large-scale single-cell datasets, where standard implementations struggle with memory usage and runtime constraints[5–10]. As a result, the performance of mixed-effects models for single-cell DE analyses has mostly been examined through simulation studies involving small numbers of subjects and cells or pseudobulk methods[2,13].

To address the challenges of large-scale scRNA-seq data analysis using linear mixed-effects models, we developed FLASH-MM, a fast and scalable LMM estimation algorithm for single-cell differential expression analysis. By leveraging summary statistics, which are pre-computed aggregate representations that capture essential information from the data without storing measurements for each individual cell, and by transferring the computation of high-dimensional matrices (number of cells) to a lower dimension (numbers of covariates and random effects) in the model estimation step, FLASH-MM achieves both computational efficiency and significantly lower memory usage. Compared to the standard LMM estimation method, lmer in the lme4 package[6], the FLASH-MM algorithm requires orders of magnitude less compute time and memory use while maintaining accuracy. We verified the accuracy, efficiency, and DE analysis performance of FLASH-MM using simulation studies. We simulated multi-subject multi-cell-type scRNA-seq datasets using real reference data based on a negative binomial (NB) distribution. We further demonstrate the application of FLASH-MM for case-control comparisons in a tuberculosis immune atlas and for cell-type-specific sex comparisons in healthy kidney data. In summary, FLASH-MM accelerates accurate single-cell differential expression analysis across diverse biological contexts, supporting the use of mixed models in large-scale, multi-subject single-cell studies.

## Results

### Overview of FLASH-MM
Single-cell RNA sequencing datasets typically consist of gene expression measurements for thousands of genes (approximately 20,000 in humans) across tens of thousands to millions of cells, often collected from multiple subjects or experimental conditions. An LMM can identify differentially expressed genes, correcting for fixed effects, modeled as covariates such as batch, sex, or treatment conditions, and subjects modeled as random effects.

We developed FLASH-MM to address the computational challenge of fitting the LMM for large-scale scRNA-seq data by efficiently estimating LMM parameters, using maximum likelihood (ML)[11] and restricted maximum likelihood (REML)[12] with a gradient descent approach (see Supplementary Information). Instead of directly processing large cell-level data matrices, our algorithm computes and operates on summary statistics, which are compact data representations, without storing information for each individual cell. Specifically, FLASH-MM operates the matrix computation by transferring the high-dimension $n \times n$ matrices (number of cells) to the low-dimension $p \times p$ and $q \times q$ matrices (numbers of fixed and random effects). This reformulation substantially reduces computational complexity from $O(mn^3)$ to $O(mn(p^2+q^2))$, and memory complexity from $O(mn)$ to $O(m\max(p, q))$, where $m$ represents the number of genes, $n$ is the number of cells and $p$ and $q$ denote the number of fixed and random effects, respectively ($n > p$ and $n > q$). By precomputing and directly using the summary statistics as inputs, FLASH-MM further reduces the computational complexity to $O(m(p^3+q^3))$, which makes LMM estimation independent of the number of cells and achieves both speed and memory efficiency (see "Methods" and Supplementary Information).

With usual LMM fitting methods, variance components are constrained to be non-negative. As a result, the asymptotic normality of the maximum likelihood estimation of variance components at the null hypothesis is invalid due to the zero variance components being on the boundary of the parameter space. Also, the asymptotic distribution of the likelihood ratio test (LRT) statistics is a mixture of Chi-squared distributions, making it more difficult to model[14,15]. Our algorithm allows variance component parameters to take negative values such that the zero variance components are no longer on the boundary of the parameter space. Thus, the usual asymptotic properties of maximum likelihood estimation at the null hypothesis remain valid under regularity conditions, which enables the use of t-statistics or z-statistics for hypothesis testing of both fixed effects and variance components, and the LRT statistics asymptotically follow a Chi-squared distribution. When the variance component parameter is positive, it suggests that the mixed-effects model is appropriately specified; otherwise, the random effect term may not be needed and should be excluded from the model.

The FLASH-MM workflow for single-cell differential expression analysis is illustrated in Fig. 1. Figure 1A shows the input structure of scRNA-seq data, represented as a log-transformed gene-by-cell count matrix, where each row corresponds to a gene's expression profile. Fixed effects can include variables such as log-library size, batch effects, biological conditions of interest, and interactions between conditions and cell types. Random effects can capture variations across subjects and correlations within subjects. Figure 1B outlines the linear mixed-effects model framework, constructed for each gene using design matrices of fixed and random effects, informed by prior knowledge of covariates and the biological question. Figure 1C illustrates the model fitting process, comprising LMM parameter estimation and hypothesis testing. Parameter estimation is performed using a gradient descent algorithm applied to summary statistics. Hypothesis testing evaluates fixed effects and their contrasts using t-statistics (see "Methods" and Supplementary Information).

### Simulation studies
We validated FLASH-MM's accuracy and computational efficiency by comparing it to the standard linear mixed model method lmer, implemented in the lme4 package[6], using simulated scRNA-seq data (See "Methods" and Supplemental Information). We also evaluated the performance of FLASH-MM in single-cell DE analysis through simulations using two key criteria: (1) control of the Type I error rate (false positive rate, FPR) and (2) statistical power (true positive rate, TPR). These metrics were compared against NEBULA[5], a generalized linear mixed-effects model (GLMM) designed for DE analysis.

### Simulating scRNA-seq data
Using PBMC 10X droplet-based scRNA-seq data from lupus patients[16] as a reference, we simulated six multi-subject multi-cell-type scRNA-seq datasets with 6000 genes and sample sizes ranging from 20,000 to 120,000 cells, increasing by 20,000 at each step, based on a negative binomial (NB) distribution. Genes were randomly selected from the reference dataset, and cells were simulated from 25 subjects across 12 cell types under two treatment conditions. Treatments, cell types, and subjects were assigned randomly with equal probability. A total of 480 differentially expressed genes were designated, specific to a cell type (Fig. S1).

### FLASH-MM has the same accuracy as lmer, but is much faster
We first validated FLASH-MM's accuracy and computational efficiency using simulated data by comparing it to the standard method, lmer, from the lme4 package[6] with its default settings. The linear mixed model (LMM) was fit to the log-transformed counts using FLASH-MM and lmer. Because lmer in the lme4 package doesn't provide p-values for hypothesis tests of coefficients, we computed p-values in this case by refitting the LMM by the lmer using the lmerTest package[17]. The differences in estimated variance components, coefficients, and p-values between FLASH-MM and lmer are shown in Fig. 2A. The model parameters (coefficients and variance components) estimated by FLASH-MM and lmer are identical up to the sixth decimal place, demonstrating the high accuracy of the FLASH-MM implementation.

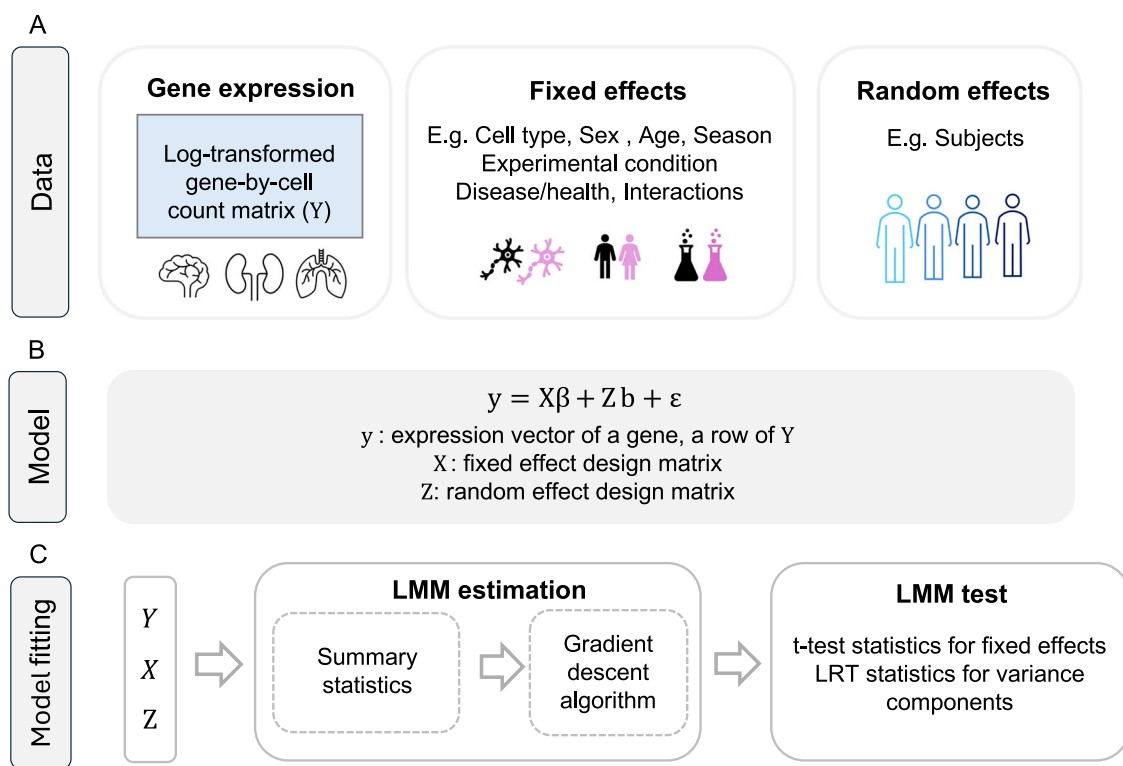

**Fig. 1 | FLASH-MM workflow for single-cell differential expression analysis.**
**A** Data: gene expression matrix **Y** = log(1 + counts), with each row corresponding to
a gene's expression profile and each column corresponding to a cell (gene-by-cell
matrix). Metadata includes various variables such as log-library size, batch effects,
biological conditions of interest, and interactions between conditions and cell
types, which could be modeled as fixed effects, and individual subjects, which
could be modeled as random effects. **B** Model: the linear mixed-effects model

(LMM) for each gene by design matrices **X** and **Z**, which are constructed based on
prior knowledge about the covariates and the biological question. **C** Model fitting:
comprises LMM estimation and tests. LMM estimation is implemented by a gra-
dient descent algorithm over summary statistics. The summary statistics are
computed as $\mathbf{X}^T\mathbf{X}$, $\mathbf{X}^T\mathbf{Y}^T$, $\mathbf{Z}^T\mathbf{X}$, $\mathbf{Z}^T\mathbf{Y}^T$, and $\mathbf{Z}^T\mathbf{Z}$. LMM tests perform hypothesis tests on
the fixed effects by *t*-test and variance components by likelihood ratio test (LRT).

FLASH-MM demonstrates substantially greater computational effi-
ciency compared to lmer with default optimizer settings, achieving a
speedup of approximately 50-fold to 140-fold as the sample size
increases from 20,000 to 120,000 cells (Fig. 2B, Table S1, measured on
a 2.8 GHz Quad-Core Intel Core i7 processor with 16 GB DDR3 RAM).
These results suggest that FLASH-MM achieves computational effi-
ciency, accuracy, and reliable inference in practice.

### FLASH-MM has a similar statistical performance to NEBULA, but is much faster

We compared the performance of FLASH-MM to NEBULA, a state-of-
the-art method for differential expression analysis of multi-subject
single-cell data based on Negative Binomial and Poisson mixed mod-
els. We ran NEBULA on our simulated data using the arguments
method = 'LN' and model = 'NBLMM', which specify the negative
binomial lognormal mixed model- the same model we used to simulate
scRNA-seq data above. We computed type I error using the simulated
non-differentially expressed (non-DE) and DE genes and compared the
power between the two methods at a sample size of 120,000 cells.

FLASH-MM effectively controls Type I error, with *p*-values
remaining within the expected range of a uniform distribution, as
shown in the quantile-quantile (QQ) plot of Fig. 2C. This expected
range is represented by 95% confidence intervals, which indicate the
natural variation we would expect if there were no true differences in
gene expression. FLASH-MM *p*-values fall within these confidence
intervals, suggesting that the method does not produce an excess of
false positives (Figs. 2C, S2). FLASH-MM demonstrates a power com-
parable to NEBULA, with a Receiver Operating Characteristic (ROC)
curve achieving an Area Under the Curve (AUC) of 0.97 for both

FLASH-MM and NEBULA (Figs. 2D, S3). The *t*-values and *p*-values cal-
culated by FLASH-MM and NEBULA demonstrate a strong correlation
(Fig. S4). However, both NEBULA and lmer required significantly
longer runtimes compared to FLASH-MM (319 and 143 times, respec-
tively, at *n* = 120,000 cells, Table S1).

We performed additional simulation studies to further evaluate
hypothesis testing performance, and showed: 1) The FLASH-MM *t*-test
for fixed effects achieves type-I error control comparable to that of the
lmerTest Satterthwaite approximation (Fig. S5); and 2) FLASH-MM's
*z*-test and LRT for variance components maintain proper type-I error
control (Fig. S6).

### FLASH-MM supports DE analysis of diverse biological scRNA-seq data

**Kidney scRNA-seq data.** We first examined the sex variation within
healthy kidney cell types using kidney scRNA-seq data[18] (Fig. 3A). The
kidney data from 19 subject samples contains 14,175 genes and 27,550
cells consisting of 19 cell types after quality control (Fig. 3B). We per-
formed a differential expression analysis using FLASH-MM to identify
the DE genes between males and females within each cell type while
considering the subjects as a random effect.

Among the various cell populations, connecting tubule (CNT)
cells have the highest number of sex-specific differentially expressed
genes (200), meeting the criteria of FDR < 0.05 and |LogFC| > 0.5
(Fig. 3C). Pathway analysis of these genes highlights distinct enrich-
ments: male CNT cells show enrichment for pathways related to acid
secretion, transporter activity, blood pressure regulation, and ion
importation, whereas female CNT cells exhibit enrichment for kinase
activity and positive regulation of receptor recycling (Fig. 3D).

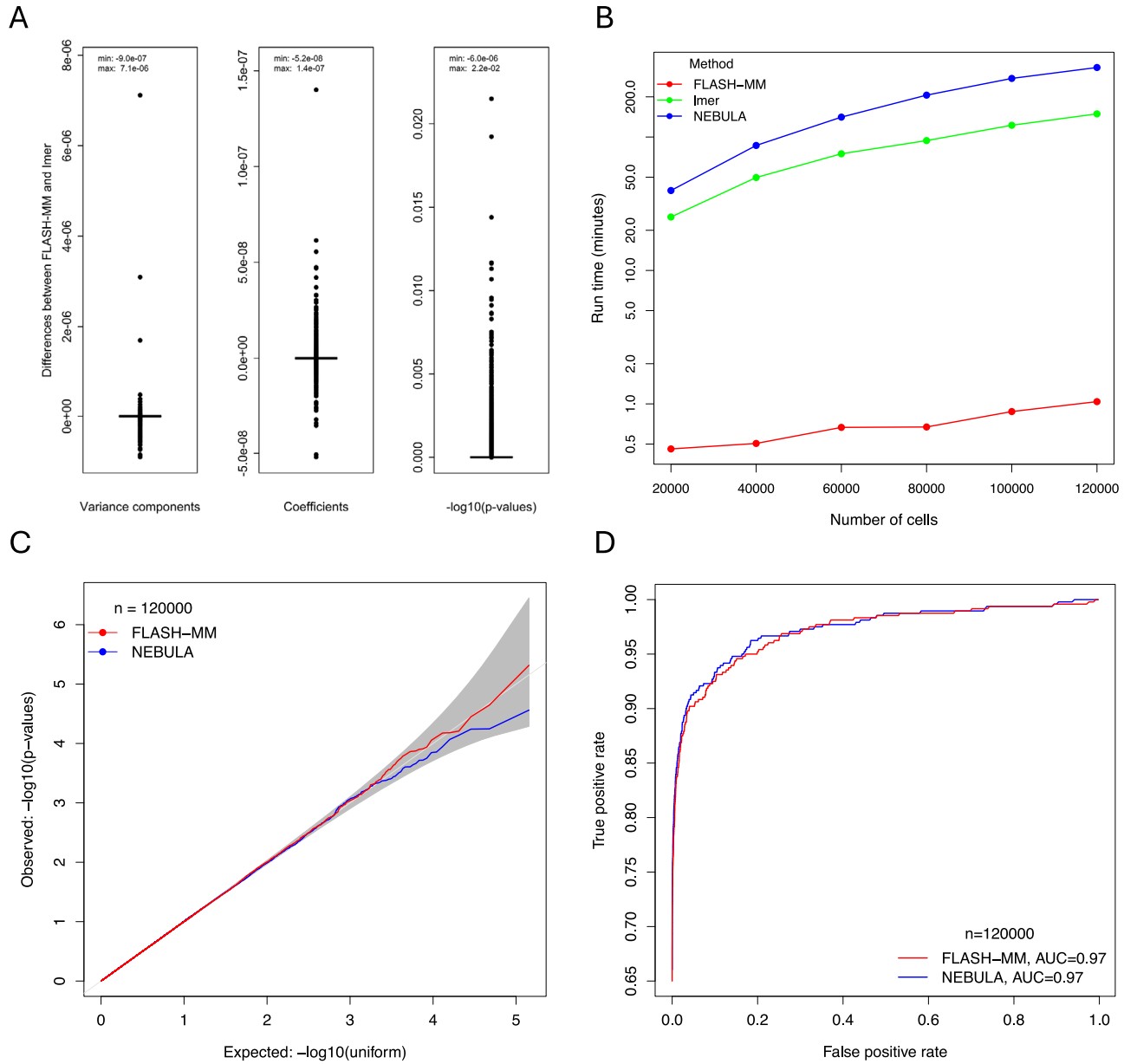

**Fig. 2 | Computational and statistical performance of FLASH-MM in differential expression analysis of simulated scRNA-seq data. A** Boxplots of differences of variance components, coefficients, and -$\log_{10}$($p$-values) between FLASH-MM and lmer fitting for each of 6000 genes across the six simulated scRNA-seq datasets. The boxplots contain 72,000 (=2*6000*6) values for variance components and 900,000 (=25*6000*6) values for coefficients and $p$-values. **B** Computation time (in minutes) for FLASH-MM, lmer, and NEBULA across the six datasets with sample sizes from 20,000 to 120,000. **C** QQ-plots of non-DE genes (negative controls), $p$-values for FLASH-MM and NEBULA. The gray area represents the 95% confidence interval, indicating the expected range under the null hypothesis. **D** ROC curves for FLASH-MM and NEBULA. Source data are provided as a Source Data file.

On the Kidney dataset, FLASH-MM required only 1.1 min of run-time, whereas *lmer* took 119.4 min under identical hardware conditions (2.8 GHz Quad-Core Intel Core i7, 16 GB DDR3 RAM).

**Tuberculosis (TB) scRNA-seq data.** We then applied FLASH-MM to single-cell transcriptomics data from 500 K memory T cells from 259 donors in a tuberculosis (TB) progression cohort[19]. After quality control, the large TB dataset contains 11,596 genes and 499,713 cells covering 29 cell states from 46 batches and 259 individual donors. We applied FLASH-MM to identify genes associated with TB status within each cellular state. FLASH-MM identified a varying number of differentially expressed genes associated with TB progression across different cell states (Fig. 4A), using a threshold of FDR < 0.05 and a positive effect size (i.e., upregulation in TB samples). The cell types with the highest number of DE genes are activated CD4+ and CD8 + T cell populations, with 1266 and 268 DE genes, respectively (Fig. 4A, B). To further investigate TB-associated signatures, we identified the top TB-enriched genes within these two cell states (Fig. 4C, D). Pathway enrichment analysis of these DE genes identifies cell-cycle pathways in activated CD4+ cells, while activated CD8+ cells show enrichment for pathways related to immune response, TCR-mediated T-cell activation, and chemokine signaling (Fig. 4E, F).

Notably, FLASH-MM completed the 500K T cell dataset analysis in 1.4 h, compared to 55.6 h (2 days and 7.6 h) for *lmer*, measured on a 2.8 GHz Quad-Core Intel Core i7 processor with 16 GB DDR3 RAM. These results demonstrate that FLASH-MM is substantially more computationally efficient and thus a more practical choice for large-scale datasets.

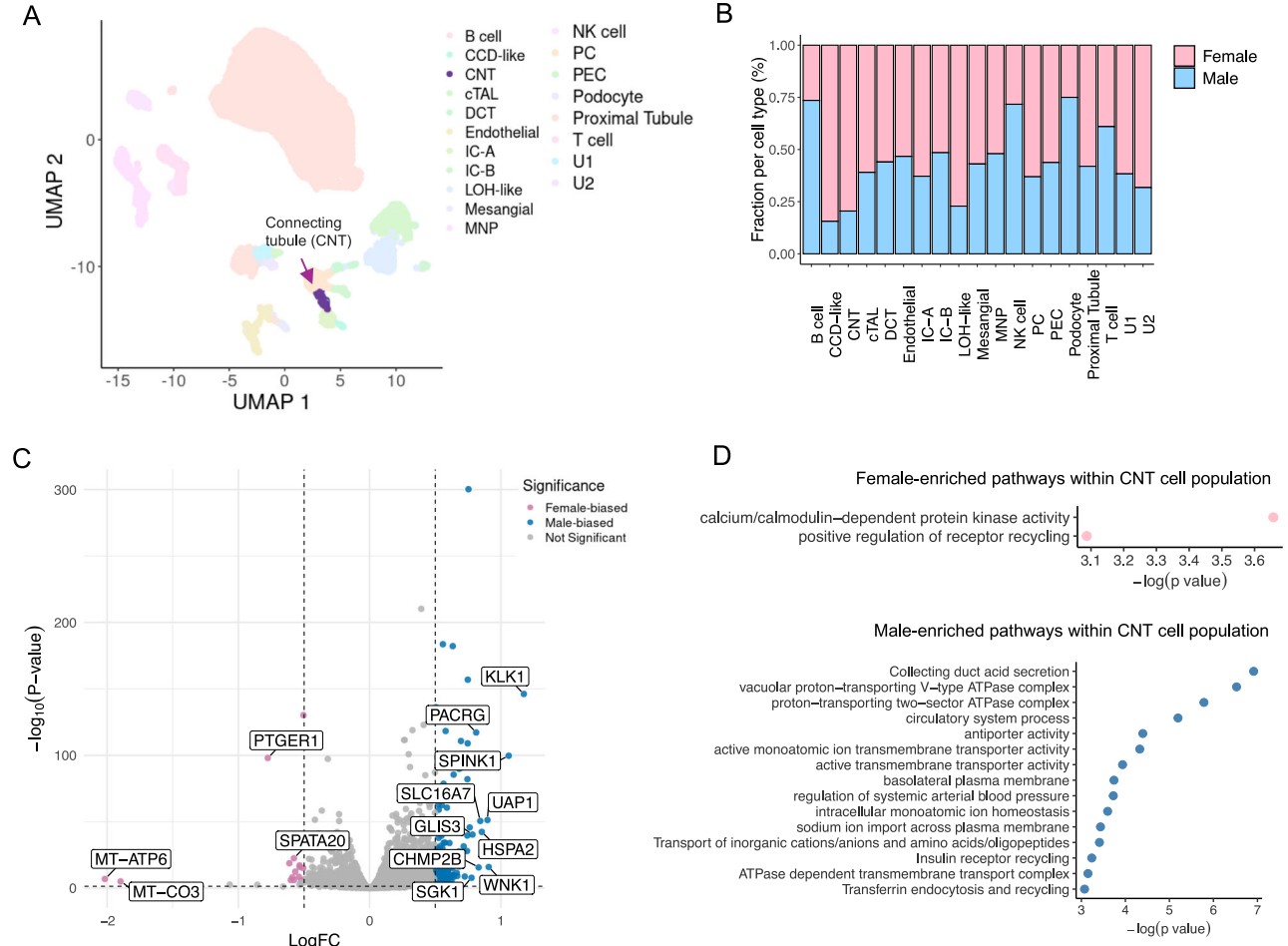

**Fig. 3 | FLASH-MM identifies sex-specific variations in a healthy human kidney map. A** UMAP projection of the healthy human kidney transcriptomic atlas, highlighting connecting tubule (CNT) cells in purple. Other cell types are shown in lighter shades for contrast. **B** Bar plot showing the proportion of male and female cells within each annotated kidney cell type. **C** Volcano plot of differential expression between sexes within the CNT cell population. Selected genes with significant male or female bias are labeled. Each point represents a gene, with the x-axis showing the sex-specific log fold change (logFC) and the y-axis showing $-\log_{10}(p\text{-value})$. Genes significantly upregulated in males or females (adjusted $p$-value < 0.05 and |logFC| > 0.5) are colored in blue and pink, respectively. A subset of the top male-biased and female-biased genes (ranked by effect size) is labeled to

minimize visual overlap. Differential expression between male and female CNT cells was tested gene-wise using a linear mixed-effects model of the form expression ~ log(library.size) + Cell_Types_Broad + Cell_Types_Broad:sex + (1|sampleID); $P$-values correspond to two-sided $t$-tests on the Cell_Types_Broad:sex interaction term and were adjusted for multiple testing across genes within the cell type using the Benjamini–Hochberg false discovery rate. **D** Pathway enrichment results for male-biased and female-biased genes within CNT cells. Dot size reflects gene set size, and x-axis position indicates significance, $-\log_{10}(p\text{-value})$. Pathway enrichment was performed separately for male-biased and female-biased gene sets using g:Profiler. Source data are provided as a Source Data file.

## Discussion

Differential expression analysis to detect changes in gene expression across conditions has long been a fundamental aspect of transcriptomics research. However, single-cell RNA sequencing data introduces unique statistical challenges, such as sample correlation, individual variation, and scalability. We developed FLASH-MM as a fast and scalable linear mixed-effects model (LMM) estimation algorithm to address these issues.

Mixed-effects models are powerful tools in single-cell studies due to their ability to model intra-subject correlations and inter-subject variabilities. Classic LMM estimation methods, like lmer in the lme4 package[6], face limitations of speed and memory use in the analysis of large-scale scRNA-seq data. These limitations have encouraged researchers to use traditional bulk RNA-seq differential expression analysis methods with pseudobulk counts by summing reads within each cell type for each subject[3,20]. While this simplifies the analysis, it sacrifices the resolution inherent in single-cell data.

To support our simulation studies, we developed a scRNA-seq simulator, implemented in the simuRNAseq function in the FLASH-MM

software distribution, to generate multi-subject multi-cell-type scRNA-seq data based on a negative binomial distribution (Supplemental information). SimuRNAseq shares similarities with muscat[20] and GLMsim[21]. Like Muscat, the simulator captures key characteristics of real single-cell RNA-seq data by modulating zero-inflation, overdispersion, variance differences, cell-level library size variation, number of clusters or cell populations, and the number of expected differentially expressed genes (Supplemental information). However, both Muscat and GLMsim have some limitations when used for scRNA-seq simulations. Muscat estimates the dispersion of the negative binomial distribution using the edgeR package, but it relies on only a subset of the reference data. As a result, it cannot scale to large scRNA-seq datasets and captures only partial information from the reference dataset. GLMsim estimates the coefficients and dispersion parameters of the NB model for each gene using glm.nb from the MASS package[22]. While effective, this approach is computationally intensive and limited to generating data of a fixed size that matches the reference data. Our scRNA-seq simulator uses the method-of-moments estimate (MME)[23] to compute dispersion parameters for the NB distribution. This

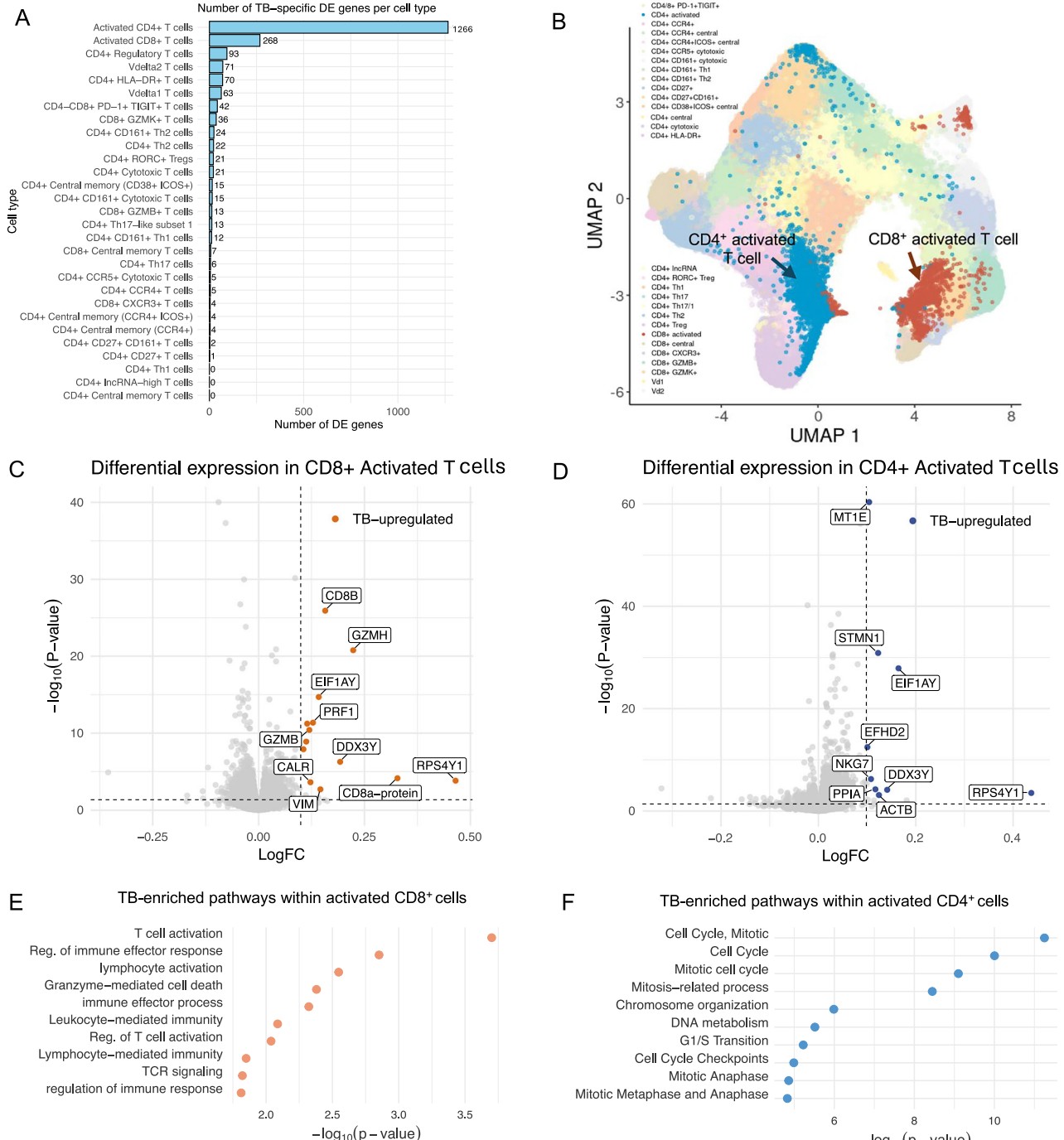

**Fig. 4 | FLASH-MM identifies TB-enriched signatures within T cell populations while accounting for confounding variables. A** Bar plot showing the number of TB-associated differentially expressed genes (FDR < 0.05 and positive effect size) identified within each T cell subtype. **B** UMAP projection of single-cell RNA-seq data from ~500 K memory T cells across 259 individuals in a TB progression cohort. CD4+-activated and CD8+-activated T cells are highlighted in blue and dark orange, respectively; all other cell types are shown in lighter tones for context. **C, D** Volcano plots of differential expression in CD8+-activated (**C**) and CD4+-activated (**D**) T cells. Each point represents a gene, with the x-axis showing the log fold change (logFC) and the y-axis showing $-\log_{10}(p\text{-value})$. Genes significantly upregulated in TB samples are colored (FDR < 0.05 and logFC > 0.1). Vertical and horizontal dashed lines indicate the logFC and p-value thresholds. A subset of the top 10 TB-upregulated genes, ranked by effect size, is labeled. Differential expression between TB and control samples in each T cell subtype was tested gene-wise using a linear mixed-effects model of the form expression ~ log(library.size) + cluster_name + cluster_name:TB_status + (1|donor); P-values correspond to two-sided t-tests on the cluster_name:TB_status interaction term and were adjusted for multiple testing across genes within each cell type using the Benjamini–Hochberg false discovery rate. **E, F** Dot plots showing enriched pathways among TB-upregulated genes in CD8+ (**E**) and CD4+ (**F**) activated T cells. x-axis position reflects enrichment significance, $-\log_{10}(p\text{-value})$. Pathway enrichment in panels E and F was performed separately for TB-upregulated gene sets using g:Profiler. Source data are provided as a Source Data file.

approach is faster, more flexible, and uses the full biological reference dataset. The performance of the scRNA-seq simulator, named simuRNAseq, is illustrated in Fig. S7.

Constructing design matrices for fixed and random effects is an important step in LMM-based DE analysis, requiring identification of the aspects of the data to be modeled, and a balance between reducing residual variance, avoiding overfitting, and managing collinearity among covariates[24]. A design matrix encodes variables such as sample conditions, batch effects, and cell types, specifying how observations are mapped to model parameters. While dataset integration methods (e.g., Harmony[25], Seurat[26]) are often applied prior to modeling, mixed-effect models can directly account for batch effects by modeling batches as fixed or random effects, depending on the study design and underlying biological question. However, in many scRNA-seq studies, samples or batches are perfectly confounded with experimental conditions. In such cases, including batch as a fixed effect may introduce collinearity and inadvertently remove the biological signal of interest. Mixed-effect models can address this issue by modeling the batch as a random effect.

For single-cell transcriptomics data, including library size as a fixed effect helps control $p$-value inflation and should generally be included in the model to help with normalization. In the analyses of the kidney and tuberculosis data, 98.8% and 99% of genes have a significant covariate of log-library size with Bonferroni correction $p$-value $< 0.05$, respectively. Based on this empirical evidence, we recommend considering including the log-library size as a covariate in the mixed-effects model. Other model design decisions should be defined by the user based on the structure of their data and the biological question under study. If samples are modeled as random effects, their number may be high (e.g., hundreds or higher), and the FLASH-MM LMM algorithm may slow down (Fig. S8). In such cases, subsampling the data or applying dimensionality reduction techniques, such as PCA, on the random effects can potentially help reduce their number to improve computational efficiency (see Supplemental Information). When the covariates of random effects are highly correlated, the dimensionality reduction technique can substantially reduce the number of random effects. Otherwise, if the covariates of random effects are independent or uncorrelated, the dimensionality reduction technique would not reduce the number of random effects.

FLASH-MM makes mixed models computationally feasible for large single-cell datasets. This scalability enables future benchmark studies to determine the optimal modeling approach in biologically relevant contexts such as subtle perturbations, continuous gradients, and rare cell populations, where accounting for cell-level variation is likely important. FLASH-MM's versatile framework can be extended to other data modalities, such as spatial transcriptomics and multiomics. Expanding its application across diverse biological data could provide opportunities for uncovering novel insights and facilitating integrated analyses in a wide range of research contexts.

## Methods
### LMM estimation and hypothesis testing
Consider the linear mixed-effects model (LMM) as expressed below[27]

$$y = X\beta + Zb + \epsilon \tag{1}$$

where $\mathbf{y}$ is an $n \times 1$ vector of observed response (expression for a gene), $\mathbf{X}$ is an $n \times p$ design matrix for fixed effects $\boldsymbol{\beta}$, $\mathbf{Z}$ is an $n \times q$ design matrix for random effects $\mathbf{b}$, and $\boldsymbol{\epsilon}$ is an $n \times 1$ vector of residual errors. The term random effects may be a combination of various random-effect components, $Zb = Z_1 b_1 + + Z_K b_K$, where $\mathbf{Z} = [Z_1, \ldots, Z_K]$, $\mathbf{b} = \begin{bmatrix} b_1^T, \ldots, b_K^T \end{bmatrix}^T$, $K$ is the number of the random-effect components, and $Z_k$ is an $n \times q_k$ design matrix for the $k$-th component. The superscript $T$ denotes the transpose of a vector or matrix. The basic assumptions are as follows: 1) The design matrix $\mathbf{X}$ is of full rank,

satisfying conditions of estimability for the parameters; 2) The random vectors $\mathbf{b}_k$ and $\boldsymbol{\epsilon}$ are independent and follow a normal distribution, $\mathbf{b}_k \sim N\left(\mathbf{0}, \sigma_k^2 \mathbf{I}_{q_k}\right)$ and $\boldsymbol{\epsilon} \sim N\left(\mathbf{0}, \sigma^2 \mathbf{I}_n\right)$. Here $\sigma_k^2$ and $\sigma^2$ are unknown parameters, called variance components, $\mathbf{0}$ is a vector or matrix of zero elements, and $\mathbf{I}_n$ is an $n \times n$ identity matrix. The random effects reflect variations between groups (subjects) and correlations within groups (subjects). Assumption (1) implies $p < n$. We also assume $q_k < n$. If $q_k > n$, we can use principal component analysis (PCA) to obtain an equivalent LMM with the number of random effects less than $n$ (see Supplementary Information).

Maximum likelihood estimation (MLE) and restricted maximum likelihood (REML) are methods for estimating fixed effects and variance components in LMMs. MLE estimates all parameters of fixed effects and variance components together, but can produce biased variance component estimates, whereas REML removes fixed effects from variance estimation, resulting in unbiased estimates. Both methods are asymptotically identical. Estimating variance components using either MLE or REML requires numerical methods, among which the iterative gradient-based methods are the most commonly used.

**Fast and scalable algorithm.** The gradient descent methods usually have a computational complexity of $O(n^3)$. We developed the summary statistics-based algorithm, FLASH-MM, to implement the gradient methods for speeding up the LMM estimation and reducing the computer memory usage. Instead of the individual cell-level data: $\mathbf{X}$, $\mathbf{Z}$ and $\mathbf{y}$, FLASH-MM uses the summary statistics: $\mathbf{X}^T \mathbf{X}$, $\mathbf{X}^T \mathbf{y}$, $\mathbf{Z}^T \mathbf{X}$, $\mathbf{Z}^T \mathbf{y}$ and $\mathbf{Z}^T \mathbf{Z}$ to estimate the LMM parameters. FLASH-MM achieves a computational complexity of $O(n(p^2 + q^2))$, which is fast and linearly scalable with the sample size $n$. These summary statistics have a low dimension and require less computer memory. By precomputing and directly using the summary statistics as inputs, the algorithm complexity is reduced to $O(p^3 + q^3)$, which makes computations independent of the sample size $n$ (number of cells) and achieves both speed and memory efficiency (see Supplementary Information section 2.1).

scRNA-seq data typically consist of gene expression measurements for thousands of genes (approximately 20,000 in humans) across thousands to millions of cells. In the single-cell differential expression analysis, the complexity of FLASH-MM is $O(mn(p^2 + q^2))$, $m$ is the number of genes, which is linearly scalable with the number of genes. By using the pre-computed summary statistics as inputs, the algorithm complexity becomes $O(m(p^3 + q^3))$.

**Hypothesis testing.** The hypothesis testing for fixed effects and variance components can be respectively defined as:

$$H_{0,i} : \beta_i = 0 \text{ versus } H_{1,i} : \beta_i \neq 0,$$

$$H_{0,k} : \sigma_k^2 = 0 \text{ versus } H_{1,k} : \sigma_k^2 > 0.$$

The variance components under the null hypothesis, $\sigma_k^2 = 0$, are on the boundary of the parameter space, in which case the MLE asymptotic normality is inappropriate. By reparameterizing the variance components, $\theta_k = \sigma_k^2 = \sigma^2 \gamma_k$, the covariance matrix, $\mathbf{V}_{\boldsymbol{\theta}} = \sigma^2 \left( \mathbf{I} + \gamma_1 \mathbf{Z}_1 \mathbf{Z}_1^T + \ldots + \gamma_K \mathbf{Z}_K \mathbf{Z}_K^T \right)$, becomes positive-definite and well-defined when $\gamma_k > -1/\lambda_{\max}$, where $\lambda_{\max} > 0$ is the largest singular value of $\mathbf{Z}\mathbf{Z}^T$ or $\mathbf{Z}^T\mathbf{Z}$. Now the parameters of variance components, $\theta_k = \sigma^2 \gamma_k$, $k = 1, \cdots, K$, can be negative. If $\theta_k > 0$, $\sigma_k^2 = \theta_k$ is definable and the mixed model is well-specified. Otherwise, the term of random effects is not needed in the model. Then the hypotheses for the variance components are extended as:

$$H_{0,k} : \theta_k \leq 0 \text{ versus } H_{1,k} : \theta_k > 0,$$

in which the zero components, $\theta_k = 0$, are no longer on the boundary of the parameter space and the MLE normal asymptotic properties hold. Then we can use the z-statistic or t-statistic for hypothesis testing of fixed effects and variance components. We can also test whether there are no random effects, that is, the variance components are equal to zero, using the likelihood ratio test (LRT) statistic. See Supplementary Information section 1.2 for details.

## Simulation methods

We generated the multi-subject multi-cell type scRNA-seq dataset by using reference data based on a negative binomial (NB) distribution. The mean of the NB distribution is taken as the sample mean for each gene. The dispersion of the NB distribution is computed by the method-of-moments estimate (MME)[23]. Compared to the maximum likelihood-based estimates, such as the glm.nb function in the MASS package[22], the MME is computationally simpler and performs reasonably well. See detailed simulation methods and performance in Supplemental Information.

## Data preprocessing and model design

In general, we selected thresholds to be in line with typical single-cell genomics projects[28], as well as to ensure there is enough data per cell and gene for the LMM to operate on. We processed the data by filtering the outliers based on: total numbers of UMI counts per cell and numbers of detected genes (cell filtering), numbers of UMI counts per gene, and UMI counts per cell (cpc) (gene filtering), which are standard in the literature[29], and described below for each dataset. Our goal was not to optimize thresholds for biological discovery, but rather to apply reasonable filters that balance noise reduction with data retention.

## Healthy human kidney scRNA-seq data

The healthy human kidney transcriptomic map[18] was generated from 27,677 cells obtained from 19 living donors (10 female and 9 male). Decontaminated raw data, processed using SoupX[30], was provided upon request by the authors. Cells were filtered based on three criteria: the number of detected features, library size, and the number of cells within each cell type (cluster).

The number of features per cell, defined as the total number of non-zero genes, was required to meet a minimum threshold of 100. Library size, calculated as the total counts per cell, was restricted to a range of 2^9 (512) to 2^16 (65,536). Cell types with fewer than 20 cells were excluded, and "Podocyte" cells were removed due to their low sample size (16 cells post filtering). Additionally, genes were filtered based on their expression levels, where the counts per cell ratio had to exceed 0.005 (i.e., the total gene count divided by the number of cells had to be greater than 0.5%). Genes were further filtered to retain those expressed in at least 16 cells, with a minimum of 10 cells in each group, total counts between 2^6 (64) and 2^20 (1,048,576), and a counts per cell ratio above the threshold. After these filtering steps, 27,550 cells and a refined set of genes were retained for downstream analyses.

The FLASH-MM model was designed to account for both technical and biological variation. The model formula used was:

~log(library.size) + Cell_Types_Broad + Cell_Types_Broad:sex + (1| sampleID),

where log-transformed library size was included as a fixed effect to normalize for differences in sequencing depth, Cell_Types_Broad captured the cell type-specific effects, and the interaction term Cell_Types_Broad:sex identified sex-specific differences within each cell type. A random effect (1|sampleID) was added to account for inter-sample variability.

## Tuberculosis (TB) T cell scRNA-seq data

The Tuberculosis (TB) memory T cell dataset was obtained from Nathan et al.[19], comprising 500,089 cells. The raw count matrix was used for preprocessing. Metadata associated with the cells includes donor identity, sex, batch, cluster annotations, and TB status. The dataset includes cells from 259 unique donors, spanning 46 batches and 29 cell clusters. In the DE analysis, we modeled the donors as a random effect and ignored the batch effect because the majority of donors were sequenced in a single batch.

Pre-processing steps consisted of removing cells and genes to remove extreme values. Library sizes were assessed using a boxplot, and cells with library sizes outside the lower whisker and above an upper threshold of 2^15 (32,768) were removed, resulting in the retention of 499,973 cells. Genes were filtered in two steps: first, genes expressed in fewer than 2^9 cells (512 cells) were excluded; second, genes with a counts-per-cell ratio less than 0.005 (i.e., total gene count divided by the number of cells below 0.5%) were removed. These filtering steps reduced the dataset to 11,596 genes and 499,973 cells, which were used for downstream analyses.

The FLASH-MM model was designed to analyze this dataset, accounting for both technical and biological variations. The model formula used was:

~log(library.size) + cluster_name + cluster_name:TB_status + (1| donor),

where log-transformed library size was normalized for differences in sequencing depth, cluster_name captured the cell type-specific effects, and the interaction term cluster_name: TB_status identified TB-associated differences within each cell type. A random effect (1|donor) was added to account for inter-donor variability.

Pathway enrichment analysis for both the kidney and TB datasets was performed using *gprofiler2*[31,32] (v0.2.3), and data visualizations were generated using *ggplot2* (v3.5.1).

## Reporting summary

Further information on research design is available in the Nature Portfolio Reporting Summary linked to this article.

## Data availability

The healthy human kidney data[18] were downloaded from the UCSC Cell Browser at https://cells.ucsc.edu/?ds=living-donor-kidney. The Tuberculosis (TB) memory T cell data[19] can be accessed from the GEO with accession code GSE158769. The stimulated PBMC data[16] was downloaded from the muscData package (Kang18_8vs8) at https://github.com/HelenaLC/muscData. Source data is provided as a Source Data file. Source data are provided with this paper.

## Code availability

The FLASH-MM software is openly available in both R and Python implementations, with example case studies, through the following repositories: R package (CRAN): https://cran.r-project.org/web/packages/FLASHMM/index.html GitHub: https://github.com/BaderLab/FLASHMM Python package (PyPI): https://pypi.org/project/FLASH-MM/ The package is distributed under the MIT License. Analysis scripts used for the case studies and data simulations are available at: https://github.com/BaderLab/FLASH-MM-analysis/. The code is archived and citable via Zenodo: https://doi.org/10.5281/zenodo.18222187.

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

## Acknowledgments

This research was supported by the Canadian Institutes for Health Research (grant PJT 469829 to GDB) and by the University of Toronto's Data Sciences Institute Doctoral Student fellowship program.

## Author contributions

Conceptualization, D.P., V.V., C.X. and G.D.B.; Methodology, C.X. and H.H.; Formal analysis, D.P. and C.X.; Writing – Original Draft, D.P. and C.X.; Writing – Review & Editing, D.P., C.X., V.V. and G.D.B.; Supervision, G.D.B.; Funding Acquisition, G.D.B.

## Competing interests

G.D.B. is on the Scientific Advisory Boards of Adela Bio and BioRender. No other competing interests are declared.
