## [Transparent Peer Review file · Nature Communications]

FLASH-MM: fast and scalable single-cell differential expression analysis using linear mixed-effects models

Corresponding Author: Professor Gary Bader

Version 0:

Reviewer comments:

Reviewer #1

(Remarks to the Author)

Single-cell RNA sequencing analysis faces a critical challenge: researchers cannot apply statistically appropriate mixed-effects models to large-scale multi-subject datasets due to computational limitations. In this study, Bader et al. develop FLASH-MM to address this challenge through an innovative statistical approach that could transform how we analyze hierarchical data structures in single-cell analysis.

Major Concerns

- 1. Statistical Framework Validation:** While the reparameterization of variance components ($\theta_k = \sigma^2\gamma_k$) is elegant and theoretically sound, the manuscript would benefit from a more thorough exploration of its implications for statistical inference. Specifically, the authors might provide empirical evidence demonstrating how this reparameterization affects the distribution of test statistics under various conditions, particularly for edge cases where variance components are near zero. This would strengthen confidence in the method's statistical properties beyond the theoretical justification currently provided.
- 2. Computational Efficiency Claims:** The computational complexity reduction is impressive, but the manuscript should more explicitly address the practical limitations regarding the number of fixed and random effects. The authors note that performance may degrade when the number of random effects is high, but do not provide clear guidelines on the practical upper limits for p and q values before computational advantages diminish. A systematic analysis showing how runtime scales with increasing numbers of fixed and random effects would provide users with valuable implementation guidance.
- 3. Data Integration Capabilities:** The manuscript demonstrates FLASH-MM's utility for analyzing individual datasets but does not adequately address its potential for integrating multiple scRNA-seq datasets from different sources or batches. Given the increasing importance of data integration in single-cell analysis, the authors should clarify whether FLASH-MM can be extended to handle batch integration problems and how the linear mixed-effects framework might facilitate multi-dataset analyses beyond what is currently possible with existing methods.
- 4. Biological Significance Assessment:** While the manuscript demonstrates FLASH-MM's application to kidney and tuberculosis datasets, it does not sufficiently explore whether the increased computational efficiency enables novel biological insights that would be unattainable with existing methods. The authors should more explicitly discuss what types of analyses are now possible that were previously infeasible, providing concrete examples of biological questions that can be addressed with FLASH-MM but not with pseudobulk or other approximation approaches.

Minor Concerns

- The preprocessing thresholds chosen for filtering cells and genes appear somewhat arbitrary. The authors should provide statistical justification for these choices or acknowledge their potential impact on results through sensitivity analyses.
- The manuscript suggests using dimensionality reduction techniques like PCA on random effects when their number is high, but does not adequately discuss how this affects statistical inference. A more detailed explanation of the trade-offs between computational efficiency and statistical accuracy would strengthen the methodology.
- The QQ plots show slight p-value deflation in NEBULA but not in FLASH-MM. This observation warrants further investigation and explanation, as it may provide insights into the comparative statistical properties of these methods.

- The discussion of FLASH-MM's simulator (simuRNAseq) focuses primarily on its computational advantages over existing simulators but lacks a validation of its biological realism. Additional comparisons demonstrating that the simulated data captures key properties of real scRNA-seq data would strengthen confidence in the simulation results.

(Remarks on code availability)

While the GitHub repository is a valuable resource, the manuscript would benefit from more detailed guidance on model specification for different experimental designs. Specific examples illustrating how to construct appropriate design matrices for common single-cell experimental scenarios would enhance usability.

Reviewer #2

(Remarks to the Author)

In their manuscript "FLASH-MM: fast and scalable single-cell differential expression analysis using linear mixed-effects models," the authors describe a new algorithm to fit the parameters of mixed-effects models efficiently. The manuscript is well written and convincingly shows that their modified inference algorithm produces parameter estimates equivalent to `lmer`, a well-established tool for this task, but is much faster. I also tested the R package installation and ran the provided example code, which worked flawlessly.

The main problem with the manuscript is the hypothesis testing, which is less carefully implemented. The authors conduct the significance testing using a Wald t-test. However, this is only appropriate for balanced and nested LMMs (<https://bbolker.github.io/mixedmodels-misc/glmmFAQ.html#what-are-the-p-values-listed-by-summaryglmerfit-etc.-are-they-reliable>), which typically do not hold for single-cell use cases described in the manuscript and could lead to anti-conservative p-values. In this context, it would also be important to rethink how the degrees of freedom are calculated. This problem is currently not discussed in the manuscript, and the package simply calculates $df = n - p$. This is not appropriate in general and leads to anti-conservative p-values. The authors need to fix the implementation (for example, implementing a likelihood ratio test with the appropriate correction for the degrees of freedom) and discuss the trade-offs between fast inference speed for the hypothesis test and potentially overconfident p-values.

In addition, there are a few minor issues with the manuscript which are listed below:

1. The authors propose a trick to avoid problems with boundary effects when conducting hypothesis tests for the variance component. The authors should provide a simulation benchmark that shows that their proposal solves the issue. In addition, the authors should carefully check the corresponding paragraphs in the Method section to make sure that all mathematical symbols are introduced and the procedure is described in sufficient detail. Furthermore, the authors should illustrate the relevance of a variance component hypothesis test for single-cell data analysis.
2. It would be useful if the authors could provide the gradients that are used to update the parameter estimates. This would also make it easier to understand why using the summary statistics instead of the raw data is permissible.
3. The authors mention that currently pseudobulking is the most popular method for differential expression analysis in single-cell data. Given the scope of the paper, it is okay that the authors do not benchmark the power and type-1 error control against pseudobulking methods. However, in that case, the authors need to rephrase the claim that pseudobulking is "problematic for comparisons involving weaker cell states", as they provide no evidence for this claim.
4. The sentence "As a result, the performance of mixed-effects models for single-cell DE analyses has mostly been examined through simulation studies [...] or pseudobulk methods" is confusing and should be rephrased.
5. The authors should also cite Muscat when referring to the original pseudobulking description in the second paragraph of the Discussion.
6. It would be helpful if the authors provided an intuitive explanation for what it means if the variance components become negative, as this would not be possible using a classical definition of variance.
7. In Figure 1A, it would be helpful to show the difference between FLASH-MM's and `lmer` log(p-values) instead or in addition to the the difference of the p-values.
8. In Figure 1B, the authors should consider showing the compute time on a log scale. In addition, it would be helpful if the authors would mention the absolute run time of their method in addition to the relative speed-up.
9. The authors write "The plot also shows a slight p-value deflation in NEBULA, meaning its p-values tend to be slightly higher than expected under the null hypothesis." The authors should adjust this claim, as the p-values from NEBULA are still inside the 95% confidence interval.
10. The authors say that "the library size should generally be included in the model". The authors should present evidence for this claim and also make sure to test that this is not just an artifact of their normalization methods ($\log(y+1)$), whereas it is more common to also adjust for the library size at this point ($\log(y/sf + 1)$, where sf are the size factors). See, for example, Ahlmann-Eltze and Huber 2023 for more details.

11. The authors write in the Discussion, that "applying dimensionality reduction [...] on the random effects can improve computational efficiency". The authors should clarify that this would not work in the most common use case, where the matrix Z is just a one-hot encoding of sample membership.

12. In Figures 3 and 4, the authors define a custom metric (score = coefficient * (-log(adjusted p-value))). This is not commonly used to combine significance and effect size in transcriptomic data analysis and is confusing. Instead, it might be better if the authors used a classical volcano plot to show both values (coefficient estimate and p-value) separately.

13. In Figures 3 and 4, the authors use bar charts to show the significance using -log(p-value) on the x-axis. This is problematic, as bar charts are interpreted as showing relative changes. It would be better to use simple dot plots.

14. The authors should fix the y-axis label in Figure 4A and give each cluster an interpretable name.

15. In the hypothesis testing section of the Methods, the authors refer to σ^2_k , without having introduced σ^2_k .

16. The authors define the thresholds used for the data analysis as powers of 2 (e.g., "total counts between 2^6 and 2^{20} "). The authors should also express those values as decimal numbers.

17. In Figure 1C in the LMM test box, the authors write "Using t-test for fixed effects and contrasts of the fixed effects". The authors should rephrase that to say that they calculate the t-statistic using a contrasts.

(Remarks on code availability)

I installed the R package and ran the example code which worked without problems.

Reviewer #3

(Remarks to the Author)

This authors describe FLASH-MM, a fast implementation of linear mixed-effects models for single-cell differential expression analysis. The authors achieve good gains in computational speed and memory efficiency by reformulating the estimation procedure to operate on summary statistics. This is a technically elegant contribution that addresses a real bottleneck in scaling mixed models in single cell data.

However, the justification for using mixed models in the first place remains unclear. Previous work has shown that pseudobulk approaches better control false discoveries and offer good control of false positives, and good reproducibility. While the authors suggest that pseudobulk methods may "sacrifice resolution," particularly for subtle or continuous effects, this claim is unsupported by data. If this is the central motivation for FLASH-MM, it must be demonstrated explicitly. To be clear it is not that I disagree that this could be true, but it is a key point to demonstrate or my recommendation to people would be to just use pseudobulk methods since these have become the de facto default for most labs.

At a minimum, the authors should benchmark FLASH-MM against pseudobulk methods on datasets with subtle or gradient-like perturbations. They should also evaluate whether the genes recovered (false positives/negatives, etc) line up with pseudobulk methods (since these have generally been validated against ground truth). Alternatively, the authors could also validate their method in settings of real data where ground truth is known. A quick test on our internal datasets revealed a surprisingly low Pearson correlation (~0.2) between FLASH-MM and pseudobulk results. Therefore, it is difficult to interpret whether FLASH-MM is recovering additional true signal or introducing noise.

Second, the manuscript claims that FLASH-MM scales to large datasets, but the validation in real data is not sufficient in my opinion. The test datasets presented here are relatively modest in size, and there is no evidence that FLASH-MM can be applied to datasets with hundreds of thousands of nuclei, which is now routine in the field. In fact, attempting to run FLASH-MM on our internal datasets with ~100,000 nuclei resulted in an immediate integer overflow (line 64 of Immfit.R). This raises concern about the robustness of the implementation under real-world conditions.

In summary, the authors present a useful computational optimization, but its value as a tool for the field in general remains unclear. To strengthen the manuscript, the authors should:

1. Provide experimental evidence that mixed models outperform pseudobulk methods (or at least match) in biologically relevant settings (e.g., subtle perturbations, gradients). This would provide a justification that the recovery of genes is real, and that then the mixed models offer an advantage in the context of complex experimental designs.
2. Demonstrate that FLASH-MM scales robustly to real-world single-cell datasets with hundreds of thousands of cells (millions would be even better), without technical failure.

(Remarks on code availability)

Version 1:

Reviewer comments:

Reviewer #1

(Remarks to the Author)

The authors address my previous concerns. I have no other suggestions.

(Remarks on code availability)

The vignette is a nice addition.

Reviewer #2

(Remarks to the Author)

After reviewing the changes the authors made, I am happy to support the publication of the manuscript.

(Remarks on code availability)

Reviewer #3

(Remarks to the Author)

My comments have been addressed. Thanks very much for the opportunity to review this great paper.

(Remarks on code availability)

REVIEWER COMMENTS

Reviewer #1 (Remarks to the Author):

Single-cell RNA sequencing analysis faces a critical challenge: researchers cannot apply statistically appropriate mixed-effects models to large-scale multi-subject datasets due to computational limitations. In this study, Bader et al. develop FLASH-MM to address this challenge through an innovative statistical approach that could transform how we analyze hierarchical data structures in single-cell analysis.

Major Concerns

1. Statistical Framework Validation: While the reparameterization of variance components ($\theta_k = \sigma^2\gamma_k$) is elegant and theoretically sound, the manuscript would benefit from a more thorough exploration of its implications for statistical inference. Specifically, the authors might provide empirical evidence demonstrating how this reparameterization affects the distribution of test statistics under various conditions, particularly for edge cases where variance components are near zero. This would strengthen confidence in the method's statistical properties beyond the theoretical justification currently provided.

Reply: We appreciate your insightful suggestion. We added simulation studies on the LMM hypothesis testing to address this (Section 3 in the supplementary material). We examined the performance of the FLASH-MM t-test for fixed effects, and the FLASH-MM z-test and likelihood ratio test (LRT) for random effects, comparing to:

- 1) the Satterthwaite approximation's t-test for fixed effects implemented in the lmerTest package¹;
- 2) lme4 anova² and the LRT in the varTestnlme package³ for testing variance components.

The lmerTest package provides tools for testing fixed effects. The varTestnlme package allows testing whether variance components are equal to zero, using the asymptotic properties of the LRT statistic (a mixture distribution).

In the simulations, we considered linear mixed models with both fixed and random slopes, and tested two scenarios: (1) the presence of a fixed slope (fixed effect), and (2) the presence of a random slope effect (zero variance). We generated unbalanced datasets with varying numbers of observations. The simulation results showed the following:

- The FLASH-MM t-test for fixed effects achieves type-I error control comparable to that of the lmerTest Satterthwaite approximation (Figure S1, reproduced below).
- FLASH-MM's z-test and LRT for variance components maintain proper type-I error control. In contrast, the lme4::anova method tends to deflate p-values, and the varTestnlme LRT also shows p-value deflation when the number of observations is small. As the number of observations increases, the performance of FLASH-MM LRT and varTestnlme LRT becomes comparable (Figure S2, reproduced below).

When the true variance components are equal to zero, which are on the boundary of the parameter space, the asymptotic distribution of the LRT statistics is a mixture of Chi-squared distributions^{4,5}. The varTestnlme package implements the LRT based on an approximate mixture distribution. The lme4 ANOVA uses Chi-squared distribution. The p-value deflation indicates that both the LRT and ANOVA deviate from the true mixture distribution. By re-parameterizing the variance components and allowing the parameters to be negative, the zero variance becomes an interior point of the parameter space. Then the asymptotic distribution of the MLE is normal, and the FLASH-MM t-test and z-test approximate the true distribution. The LRT statistics asymptotically follow a Chi-squared distribution.

Figure S1

Figure S2

2. Computational Efficiency Claims: The computational complexity reduction is impressive, but the manuscript should more explicitly address the practical limitations regarding the number of fixed and random effects. The authors note that performance may degrade when the number of random effects is high, but do not provide clear guidelines on the practical upper limits for p and q values before computational advantages diminish. A systematic analysis showing how runtime scales with increasing numbers of fixed and random effects would provide users with valuable implementation guidance.

Reply: By precomputing and directly using the summary statistics as inputs, FLASH-MM has a computational complexity $O(m(p^3 + q^3))$, where m is the number of genes, p and q are the numbers of fixed and random effects, respectively. When $q < 100$, the FLASH-MM runtime is

under a few minutes. When q equals to hundreds, the FLASH-MM runtime would take about hours for a large-scale dataset, but it is still much faster than lmer. For example, in the Tuberculosis (TB) scRNA-seq data analysis, where $q = 259$, FLASH-MM took about 1.4 hours of runtime, whereas lmer in lme4 spent 55.6 hours (2 days and 7.6 hours). All the runtimes were measured on a 2.8 GHz Quad-Core Intel Core i7 processor with 16 GB DDR3 RAM.

We added simulation analysis showing how runtime changes with the number of random effects. We generated data containing 100,000 cells and 6,000 genes with 16 cell-types and number of individuals (i.e., the number of random effects) ranging from 50 to 400. The data were fit by the lmmfit function in the FLASHMM package. The runtimes are shown below (now added as Extended Figure 4). The empirically measured cubic scaling matches our theoretical cubic scaling, in this case with the q (number of random effects) parameter.

3. Data Integration Capabilities: *The manuscript demonstrates FLASH-MM's utility for analyzing individual datasets but does not adequately address its potential for integrating multiple scRNA-seq datasets from different sources or batches. Given the increasing importance of data integration in single-cell analysis, the authors should clarify whether FLASH-MM can be extended to handle batch integration problems and how the linear mixed-effects framework might facilitate multi-dataset analyses beyond what is currently possible with existing methods.*

Reply: Thanks for this question. Existing integration methods, such as Harmony or those in Seurat, can be used before fitting mixed-effect models. Alternatively, similar to a general linear model design, the batch can be modelled as a fixed effect, based on the study design and underlying biological question. In many scRNA-seq studies, samples or batches, such as donors, are perfectly correlated with experimental conditions. In such cases, including the batches as fixed effects will lead to collinearity and also regress out the biological signal of interest. In this case, the mixed-effects model can handle the batches via random effects. We have updated the manuscript discussion and the R package vignette to address this point.

Paragraph added: *“While dataset integration methods (e.g., Harmony⁶, Seurat⁷) are often applied prior to modeling, mixed-effect models can directly account for batch effects by modeling batches as fixed or random effects, depending on the study design and underlying biological question. In many scRNA-seq studies, samples or batches are perfectly confounded with experimental conditions. In such cases, including batch as a fixed effect may introduce collinearity and inadvertently remove the biological signal of interest. Mixed-effect models can address this issue by modeling the batch as a random effect.”*

4. Biological Significance Assessment: While the manuscript demonstrates FLASH-MM's application to kidney and tuberculosis datasets, it does not sufficiently explore whether the increased computational efficiency enables novel biological insights that would be unattainable with existing methods. The authors should more explicitly discuss what types of analyses are now possible that were previously infeasible, providing concrete examples of biological questions that can be addressed with FLASH-MM but not with pseudobulk or other approximation approaches.

Reply: We agree that FLASH-MM enables a broader range of biologically motivated analyses by making mixed models tractable at single-cell resolution. Novel biological insights are likely to become attainable because larger datasets can now be analyzed using the LMM method, leading to increased sensitivity. We have clarified this point in the manuscript's discussion. We do not claim novel findings, as these would require additional evidence, such as from new experiments, to validate.

Minor Concerns

- *The preprocessing thresholds chosen for filtering cells and genes appear somewhat arbitrary. The authors should provide statistical justification for these choices or acknowledge their potential impact on results through sensitivity analyses.*

Reply: In general, we selected thresholds to be in line with typical single cell genomics projects⁸, as well as to ensure there is enough data per cell and gene for the LMM to operate on. We processed the data by filtering the outliers based on: total numbers of UMI counts per cell and numbers of detected genes (cell filtering), numbers of UMI counts per gene, and UMI counts per cell (cpc) (gene filtering), which are described by the paper: “Luecken et al.

Current best practices in single-cell RNA-seq analysis: A tutorial. *Mol Syst Biol.* 2019”⁹, and Seurat tutorials: https://satijalab.org/seurat/articles/pbmc3k_tutorial.html

To help evaluate the importance of these parameters, we performed a sensitivity analysis to assess the robustness of FLASH-MM results on the kidney dataset to the filtering thresholds. Specifically, we repeated the analysis under both more lenient and more stringent gene filtering criteria and compared effect size estimates for a representative contrast in the mixed model (connecting tubule (CNT) cells:Male).

- Default thresholds: required UMI detection in at least 16 cells overall and 10 cells per group to ensure sufficient coverage for comparison across conditions, a minimum of 64 total UMI counts to reduce noise from low-expression genes, a maximum of 1 million counts to avoid bias from extreme outliers, and a minimum average expression (CPC > 0.005) to filter out genes that contribute negligibly to the transcriptome.
- Lenient thresholds: required only 8 cells of detection overall and 5 cells per group, with a minimum expression threshold of 32 counts and 0.25% average expression per cell.
- Stringent thresholds: required 32 cells overall and 15 per group, a minimum of 128 counts, and 1% average expression.

As shown below, the results are nearly identical, with Pearson correlations above 0.999 across all comparisons:

- $\text{cor}(\text{default}, \text{lenient}) = 0.9999977$
- $\text{cor}(\text{default}, \text{stringent}) = 0.9999956$
- $\text{cor}(\text{lenient}, \text{stringent}) = 0.9999976$

These findings support the idea that FLASH-MM’s inferences are robust to reasonable variation in filtering parameters.

- *The manuscript suggests using dimensionality reduction techniques like PCA on random effects when their number is high, but does not adequately discuss how this affects statistical inference. A more detailed explanation of the trade-offs between computational efficiency and statistical accuracy would strengthen the methodology.*

Reply: When the number of random effects is large, (e.g., >500), dimensionality reduction techniques such as principal component analysis can be applied to reduce the number of random effects. It does not sacrifice accuracy from a theoretical perspective as the reduced model is equivalent to the original one, only the number of random effects is reduced (See Supplementary Section 2.2 LMM with dimension reduction).

When the covariates associated with the random effects are correlated, principal component analysis can substantially reduce dimensionality without sacrificing accuracy. However, if the covariates of random effects are independent or uncorrelated, the dimensionality reduction would offer no advantage. For this reason, principal component analysis is recommended as an optional way to speed up the LMM fitting in the case of having a large (e.g., >500) number of random effects. We have included a more detailed explanation of this in the revised manuscript's discussion section.

- *The QQ plots show slight p-value deflation in NEBULA but not in FLASH-MM. This observation warrants further investigation and explanation, as it may provide insights into the comparative statistical properties of these methods.*

Reply: The observed p-value deflation is minimal and remains within the boundaries of the 95% confidence interval. Accordingly, we have removed the corresponding sentence from the manuscript to clarify our interpretation.

- *The discussion of FLASH-MM's simulator (simuRNAseq) focuses primarily on its computational advantages over existing simulators but lacks a validation of its biological realism. Additional comparisons demonstrating that the simulated data captures key properties of real scRNA-seq data would strengthen confidence in the simulation results.*

Reply: We apologize for the ambiguity in the original manuscript. The key points are clarified below. The simuRNAseq simulator was developed to support our benchmarking analyses of the FLASH-MM method. The simulator shares similarities with muscat¹⁰ and GLMsim¹¹. The main distinction is that we estimate the dispersion parameter of the negative binomial distribution using a method-of-moments estimate (MME)¹², as opposed to the maximum likelihood estimate (MLE) used in muscat and GLMsim. The MME is computationally simple and fast, and provides dispersion estimates consistent with those from the MLE (Supplemental Figure S4a).

Like muscat, the simulator captures key characteristics of real single-cell RNA-seq data by modulating zero-inflation, overdispersion, variance differences, cell-level library size variation, number of clusters or cell populations, and the number of expected differentially expressed genes. Supplemental Figure S4a shows that MME and MLE yield consistent dispersion estimates. Figures S4b–d further demonstrate that the simulated data recapitulate the distributions of counts, variances, and library sizes observed in real scRNA-seq datasets. We have now further clarified this in Sections 4.1 and 4.2 of Supplementary Material.

Reviewer #1 (Remarks on code availability):

While the GitHub repository is a valuable resource, the manuscript would benefit from more detailed guidance on model specification for different experimental designs. Specific examples illustrating how to construct appropriate design matrices for common single-cell experimental scenarios would enhance usability.

Reply: Thanks for the suggestion. A detailed tutorial has been added to the package vignette.

Reviewer #2 (Remarks to the Author):

In their manuscript "FLASH-MM: fast and scalable single-cell differential expression analysis using linear mixed-effects models," the authors describe a new algorithm to fit the parameters of mixed-effects models efficiently. The manuscript is well written and convincingly shows that their modified inference algorithm produces parameter estimates equivalent to `lmer`, a well-established tool for this task, but is much faster. I also tested the R package installation and ran the provided example code, which worked flawlessly.

The main problem with the manuscript is the hypothesis testing, which is less carefully implemented. The authors conduct the significance testing using a Wald t-test. However, this is only appropriate for balanced and nested LMMs (<https://bbolker.github.io/mixedmodels-misc/glmmFAQ.html#what-are-the-p-values-listed-by-summaryglmerfit-etc.-are-they-reliable>), which typically do not hold for single-cell use cases described in the manuscript and could lead to anti-conservative p-values.

In this context, it would also be important to rethink how the degrees of freedom are calculated. This problem is currently not discussed in the manuscript, and the package simply calculates $df = n - p$ [14] [15]. This is not appropriate in general and leads to anti-conservative p-values. The authors need to fix the implementation (for example, implementing a likelihood ratio test with the appropriate correction for the degrees of freedom) and discuss the trade-offs between fast inference speed for the hypothesis test and potentially overconfident p-values.

Reply: Thank you for raising the important issue of hypothesis testing, which remains one of the most challenging aspects of linear mixed models, especially in the context of testing

variance components. The main difficulty arises because zero variance lies on the boundary of the parameter space. In this case, the asymptotic distribution of the likelihood ratio test (LRT) statistic under the null is no longer a chi-squared distribution but a mixture of chi-squared distributions^{4,5}, which leads to complexities in calculating the distribution.

In FLASH-MM, we take a different approach by reparameterizing the model such that variance components are allowed to take negative values. This shifts the boundary, making zero variance an interior point of the parameter space. As a result, the asymptotic distribution of the maximum likelihood estimator (MLE) under the null becomes approximately normal, allowing us to use Wald-type tests (t- or z-tests) without requiring special adjustments for degrees of freedom. We also added the LRT statistic for testing whether there are random effects, that is, the variance components are equal to zero.

Our simulations for scRNA-seq differential expression analysis show that the Wald t-test for fixed effects achieves good type-I error control (Figure 2C). Following your suggestion and that of Reviewer 1, we have now included additional simulation studies to further evaluate hypothesis testing performance (see our reply to Reviewer 1). The simulation results showed:

- The FLASH-MM t-test for fixed effects achieves type-I error control comparable to that of the lmerTest Satterthwaite approximation (Figure S1).
- FLASH-MM's z-test and LRT for variance components maintain proper type-I error control. In contrast, the lme4::anova method tends to deflate p-values, and the varTestnlme LRT also shows p-value deflation when the number of observations is small. As the number of observations increases, the performance of FLASH-MM LRT and varTestnlme LRT becomes comparable (Figure S2).

We have added details about this to Section 3 of the Supplementary Material.

To address the last part of the comment:

“discuss the trade-offs between fast inference speed for the hypothesis test and potentially overconfident p-values”

Reply: Our model achieves computational efficiency without sacrificing statistical accuracy. We compared FLASH-MM to the standard implementation in the lme4 package and the lmerTest package, which builds on lme4 and provides p-values for fixed effects using the Satterthwaite approximation for degrees of freedom. Despite FLASH-MM's speed advantages, we observed extremely close agreement between the two methods, with accuracy matching up to the sixth decimal place (Figures 2A and 2B). These results suggest that FLASH-MM achieves both computational efficiency and reliable inference in practice. We added a note about this in the result section: “These results suggest that FLASH-MM achieves both computational efficiency and reliable inference in practice”.

In addition, there are a few minor issues with the manuscript which are listed below:

1. The authors propose a trick to avoid problems with boundary effects when conducting hypothesis tests for the variance component. The authors should provide a simulation benchmark that shows that their proposal solves the issue.

Reply: We appreciate your suggestion. As mentioned above and now in Section 3 in the supplement, we added simulation studies on hypothesis testing, in which we studied two cases: (1) testing fixed effects by the t-test and (2) testing variance components by the z-test and LRT. Simulation results indicate that FLASH-MM has good Type-I error control in both cases.

In addition, the authors should carefully check the corresponding paragraphs in the Method section to make sure that all mathematical symbols are introduced and the procedure is described in sufficient detail.

Reply: Thank you for pointing this out. We have carefully reviewed the Methods section to ensure that all mathematical symbols are clearly introduced and that the procedures are described in sufficient detail.

Furthermore, the authors should illustrate the relevance of a variance component hypothesis test for single-cell data analysis.

Reply: A variance component hypothesis test assesses whether the inclusion of a random effect (e.g., sample or donor) significantly improves model fit. In the context of single-cell data, a significantly positive variance component indicates that modeling between-sample variation is necessary, supporting the use of mixed-effects models. Conversely, a non-significant or near-zero variance suggests that the random effect may not be included. We've now added this explanation under Overview of FLASH-MM: "When the variance component parameter is positive, it suggests that the mixed-effects model is appropriately specified; otherwise, the random effect term may not be needed and should be excluded from the model."

2. It would be useful if the authors could provide the gradients that are used to update the parameter estimates. This would also make it easier to understand why using the summary statistics instead of the raw data is permissible.

Reply: We added the formulas for computing the gradients, see equations (46) in the supplementary materials, from which it is clear that the gradients can be computed using the summary statistics.

3. The authors mention that currently pseudobulking is the most popular method for differential expression analysis in single-cell data. Given the scope of the paper, it is okay that the authors do not benchmark the power and type-1 error control against pseudobulking

methods. However, in that case, the authors need to rephrase the claim that pseudobulking is "problematic for comparisons involving weaker cell states", as they provide no evidence for this claim.

Reply: Thank you for pointing this out. Since it is not the focus of our work, we have removed the sentence referring to pseudobulking as “problematic for comparisons involving weaker cell states” from the manuscript.

4. The sentence "As a result, the performance of mixed-effects models for single-cell DE analyses has mostly been examined through simulation studies [...] or pseudobulk methods" is confusing and should be rephrased.

Reply: We have rephrased the sentence.

“As a result, the performance of mixed-effects models for single-cell DE analyses has mostly been examined through simulation studies involving small numbers of subjects and cells or pseudobulk methods”

5. The authors should also cite Muscat when referring to the original pseudobulking description in the second paragraph of the Discussion.

Reply: We added the citation.

6. It would be helpful if the authors provided an intuitive explanation for what it means if the variance components become negative, as this would not be possible using a classical definition of variance.

Reply: Thanks for the suggestion. We've added an intuitive explanation under Overview of FLASH-MM: “When the variance component parameter is positive, it suggests that the mixed-effects model is appropriately specified; otherwise, the random effect term may not be needed and should be excluded from the model.”

7. In Figure 1A, it would be helpful to show the difference between FLASH-MM's and Imer log(p-values) instead or in addition to the difference of the p-values.

Reply: Thanks for the suggestion. The third boxplot in Figure 2A was updated to show log(p-values), and the figure has been split into three separate panels to improve clarity.

8. In Figure 1B, the authors should consider showing the compute time on a log scale. In addition, it would be helpful if the authors would mention the absolute run time of their method in addition to the relative speed-up.

Reply: Figure 2B was changed to use a log scale. The absolute run time table is included in Supplementary Table S1.

9. The authors write "The plot also shows a slight p-value deflation in NEBULA, meaning its p-values tend to be slightly higher than expected under the null hypothesis." The authors

should adjust this claim, as the p-values from NEBULA are still inside the 95% confidence interval.

Reply: We agree with the reviewer's comment that the p-values from NEBULA are still inside the 95% confidence interval and we have removed the corresponding sentence from the manuscript.

10. The authors say that "the library size should generally be included in the model". The authors should present evidence for this claim and also make sure to test that this is not just an artifact of their normalization methods ($\log(y+1)$), whereas it is more common to also adjust for the library size at this point ($\log(y/sf + 1)$, where sf are the size factors). See, for example, Ahlmann-Eltze and Huber 2023 for more details.

Reply: Thank you for the helpful comment and the reference. We agree that if the expression data is already normalized, the log-library size should not be included in the model. For the log-transformed raw counts data, we found that the covariate of log-library size is highly significant in the simulation studies and real biological data analyses. In the analyses of the kidney and tuberculosis data, 98.8% and 99% of genes have a significant covariate of log-library size with Bonferroni correction p-value < 0.05 , respectively. Based on the empirical evidence, we recommend including the log-library size as a covariate in the mixed-effects model. We revised the relevant sentences in the Discussion section for clarity: "Based on the empirical evidence, we recommend considering including the log-library size as a covariate in the mixed-effects model."

11. The authors write in the Discussion, that "applying dimensionality reduction [...] on the random effects can improve computational efficiency". The authors should clarify that this would not work in the most common use case, where the matrix Z is just a one-hot encoding of sample membership.

Reply: Thank you for this comment. You are correct that dimensionality reduction is not applicable when the random effects design matrix Z is a one-hot encoding of sample membership, as the columns are orthogonal and uncorrelated. We have revised the Discussion section to clarify this point by adding the following sentences: "When the covariates of random effects are highly correlated, the dimensionality reduction technique can substantially reduce the number of random effects. Otherwise, if the covariates of random effects are independent or uncorrelated, the dimensionality reduction technique would not reduce the number of random effects."

*12. In Figures 3 and 4, the authors define a custom metric (score = coefficient * (-log(adjusted p-value))). This is not commonly used to combine significance and effect size in transcriptomic data analysis and is confusing. Instead, it might be better if the authors used a classical volcano plot to show both values (coefficient estimate and p-value) separately.*

Reply: We have replaced the custom metric with the standard volcano plots in the updated version of the main figures 3 and 4.

13. In Figures 3 and 4, the authors use bar charts to show the significance using $-\log(p\text{-value})$ on the x-axis. This is problematic, as bar charts are interpreted as showing relative changes. It would be better to use simple dot plots.

Reply: The bar charts are now replaced with dot plots.

14. The authors should fix the y-axis label in Figure 4A and give each cluster an interpretable name.

Reply: Done.

15. In the hypothesis testing section of the Methods, the authors refer to σ^2_k , without having introduced σ^2_k .

Reply: We previously introduced the notation σ^2_i but later referred to it as σ^2_k . We apologize for the inconsistency; this has been corrected in the Methods section.

16. The authors define the thresholds used for the data analysis as powers of 2 (e.g., "total counts between 2^6 and 2^{20} "). The authors should also express those values as decimal numbers.

Reply: Done.

17. In Figure 1C in the LMM test box, the authors write "Using t-test for fixed effects and contrasts of the fixed effects". The authors should rephrase that to say that they calculate the t-statistic using a contrasts.

Reply: Changed to "t-test statistics for fixed effects. LRT statistics for variance components."

Reviewer #2 (Remarks on code availability):

I installed the R package and ran the example code which worked without problems.

Reviewer #3 (Remarks to the Author):

The authors describe FLASH-MM, a fast implementation of linear mixed-effects models for single-cell differential expression analysis. The authors achieve good gains in computational speed and memory efficiency by reformulating the estimation procedure to operate on summary statistics. This is a technically elegant contribution that addresses a real bottleneck in scaling mixed models in single cell data.

However, the justification for using mixed models in the first place remains unclear. Previous work has shown that pseudobulk approaches better control false discoveries and offer good control of false positives, and good reproducibility. While the authors suggest that pseudobulk

methods may “sacrifice resolution,” particularly for subtle or continuous effects, this claim is unsupported by data. If this is the central motivation for FLASH-MM, it must be demonstrated explicitly. To be clear it is not that I disagree that this could be true, but it is a key point to demonstrate or my recommendation to people would be to just use pseudobulk methods since these have become the de facto default for most labs.

At a minimum, the authors should benchmark FLASH-MM against pseudobulk methods on datasets with subtle or gradient-like perturbations. They should also evaluate whether the genes recovered (false positives/negatives, etc) line up with pseudobulk methods (since these have generally been validated against ground truth). Alternatively, the authors could also validate their method in settings of real data where ground truth is known. A quick test on our internal datasets revealed a surprisingly low Pearson correlation (~0.2) between FLASH-MM and pseudobulk results. Therefore, it is difficult to interpret whether FLASH-MM is recovering additional true signal or introducing noise.

Second, the manuscript claims that FLASH-MM scales to large datasets, but the validation in real data is not sufficient in my opinion. The test datasets presented here are relatively modest in size, and there is no evidence that FLASH-MM can be applied to datasets with hundreds of thousands of nuclei, which is now routine in the field. In fact, attempting to run FLASH-MM on our internal datasets with ~100,000 nuclei resulted in an immediate integer overflow (line 64 of Immfit.R). This raises concern about the robustness of the implementation under real-world conditions.

In summary, the authors present a useful computational optimization, but its value as a tool for the field in general remains unclear. To strengthen the manuscript, the authors should:

- 1. Provide experimental evidence that mixed models outperform pseudobulk methods (or at least match) in biologically relevant settings (e.g., subtle perturbations, gradients). This would provide a justification that the recovery of genes is real, and that then the mixed models offer an advantage in the context of complex experimental designs.*

Reply: Thank you for your insightful comment. As our main aim is to describe a faster LMM and not to compare it with pseudobulk, we removed statements about pseudobulk comparisons from the paper.

The relative merits of pseudobulk and mixed model approaches for single-cell RNA-seq differential expression analysis remain an area of active debate (summarized below) and should be pursued as a dedicated future study, which could be aided by FLASH-MM. Early work by Zimmerman et al. (2021)¹³ argued that pseudobulk methods suffer from pseudoreplication, underestimating variance by treating cells as independent observations. They advocated for mixed models that explicitly account for sample-level correlation. This view was later challenged by Murphy et al. (2022)¹⁴, who showed that pseudobulk methods can outperform mixed models under certain metrics. More recently, Gagnon et al. (2022)¹⁵

benchmarked twelve differential expression methods across multiple metrics and simulation settings, finding that cell-level approaches such as NEBULA¹⁶ and glmmTMB¹⁷ outperformed pseudobulk methods overall.

Intuitively, the pseudobulk approach has the following drawbacks: (1) It ignores the variation between subjects. The variances of pseudocounts can differ across subjects, but pseudobulk methods assume homogeneous variance across individuals. (2) It reduces statistical power by collapsing cell-level data; therefore, lowering the sample size to the number of subjects. These methods substantially lose power in studies with few individuals.

However, the relative importance of these may be dataset dependent, which is another reason a more in depth study should be performed in the future.

2. Demonstrate that FLASH-MM scales robustly to real-world single-cell datasets with hundreds of thousands of cells (millions would be even better), without technical failure.

Reply: Thank you for raising the technical issue regarding the integer overflow problem that you met when running FLASH-MM. The problem came from computation of argument, $nBlocks = \text{ceiling}(nrow(Y) * ncol(Y) * 1e-8)$, in the “lmmfit” function, where $nrow(Y) * ncol(Y)$ can cause an integer overflow for large data. We have fixed it by changing as $nBlocks = \text{ceiling}((ncol(Y) * 1e-08) * nrow(Y))$.

We test this by successfully fitting simulated data with 1,000,000 cells and 5,000 genes. However, for an extreme large-scale dataset (e.g., millions of cells), we recommend to pre-compute and store the summary statistics, and then reload the summary statistics and run “lmm” function instead of “lmmfit” function. This is explained on GitHub and the vignette page for users:

<https://cran.r-project.org/web/packages/FLASHMM/vignettes/FLASHMM-vignette.html>

<https://github.com/BaderLab/FLASHMM/tree/main>

References

1. Kuznetsova, A., Brockhoff, P. B. & Christensen, R. H. B. lmerTest package: tests in linear mixed effects models. *J. Stat. Softw.* **82**, 1–26 (2017).
2. Bates, D., Mächler, M., Bolker, B. & Walker, S. Fitting linear mixed-effects models using lme4. *J. Stat. Softw.* **67**, 1–48 (2015).
3. Baey, C. & Kuhn, E. varTestnlme : An R Package for Variance Components Testing in Linear and Nonlinear Mixed-Effects Models. *J. Stat. Softw.* **107**, (2023).
4. Self, S. G. & Liang, K.-Y. Asymptotic properties of maximum likelihood estimators and likelihood ratio tests under nonstandard conditions. *J. Am. Stat. Assoc.* **82**, 605 (1987).
5. Stram, D. O. & Lee, J. W. Variance components testing in the longitudinal mixed effects model. *Biometrics* **50**, 1171–1177 (1994).
6. Korsunsky, I. *et al.* Fast, sensitive and accurate integration of single-cell data with

- Harmony. *Nat. Methods* **16**, 1289–1296 (2019).
7. Stuart, T. *et al.* Comprehensive Integration of Single-Cell Data. *Cell* **177**, 1888-1902.e21 (2019).
 8. Heumos, L. *et al.* Best practices for single-cell analysis across modalities. *Nat. Rev. Genet.* **24**, 550–572 (2023).
 9. Luecken, M. D. & Theis, F. J. Current best practices in single-cell RNA-seq analysis: a tutorial. *Mol. Syst. Biol.* **15**, e8746 (2019).
 10. Crowell, H. L. *et al.* muscat detects subpopulation-specific state transitions from multi-sample multi-condition single-cell transcriptomics data. *Nat. Commun.* **11**, 6077 (2020).
 11. Wang, J., Chen, L., Thijssen, R., Phipson, B. & Speed, T. P. GLMsim: a GLM-based single cell RNA-seq simulator incorporating batch and biological effects. *BioRxiv* (2024) doi:10.1101/2024.03.20.586030.
 12. Clark, S. J. & Perry, J. N. Estimation of the Negative Binomial Parameter κ by Maximum Quasi-Likelihood. *Biometrics* **45**, 309 (1989).
 13. Zimmerman, K. D., Espeland, M. A. & Langefeld, C. D. A practical solution to pseudoreplication bias in single-cell studies. *Nat. Commun.* **12**, 738 (2021).
 14. Murphy, A. E. & Skene, N. G. A balanced measure shows superior performance of pseudobulk methods in single-cell RNA-sequencing analysis. *Nat. Commun.* **13**, 7851 (2022).
 15. Gagnon, J. *et al.* Recommendations of scRNA-seq Differential Gene Expression Analysis Based on Comprehensive Benchmarking. *Life (Basel)* **12**, (2022).
 16. He, L. *et al.* NEBULA is a fast negative binomial mixed model for differential or co-expression analysis of large-scale multi-subject single-cell data. *Commun. Biol.* **4**, 629 (2021).
 17. Brooks, M., E. *et al.* glmmTMB Balances Speed and Flexibility Among Packages for Zero-inflated Generalized Linear Mixed Modeling. *R J.* **9**, 378 (2017).